# Dialysate and plasma meropenem concentrations in continuous intraperitoneal regimen during peritoneal-dialysis-related peritonitis

**Thatsaphan Srithongkul**[1], **Sukit Raksasuk**[1], **Bulaporn Techajongnumchai**[1], **Suchai Sritippayawan**[1], **Pornpan Koomanachai**[2]*

1 Division of Nephrology, Department of Medicine, Faculty of Medicine, Siriraj Hospital, Mahidol University, Bangkok, Thailand, 2 Division of Infectious Diseases and Tropical Medicine, Department of Medicine, Faculty of Medicine, Siriraj Hospital, Mahidol University, Bangkok, Thailand

* nokmed@yahoo.com

## Abstract

### Background

A single dose of intraperitoneal (IP) meropenem is recommended for peritoneal-dialysis (PD)-related peritonitis stemming from extended-spectrum β-lactamase-producing organisms. However, data on IP meropenem is limited.

### Methods

This prospective, descriptive study was conducted to examine plasma and dialysate meropenem levels during continuous IP meropenem administration in five patients with PD-related peritonitis. All patients received an IP meropenem loading dose of 500 mg, followed by IP meropenem at 125 mg/L, with four exchanges daily. The plasma and dialysate meropenem concentrations were measured at specified intervals for a 24-hour period utilizing a high-performance, liquid chromatography method.

### Results

Five patients with PD related peritonitis were studied. The mean-maximum dialysate and plasma meropenem levels were 158.1 mg/L (standard deviation [SD] ± 62.9) and 29.4 mg/L (SD ± 15.9), respectively. The mean dialysate meropenem level was at its minimum of 32.6 mg/L (SD ± 19.1) at 24 hours. Throughout the period, the dialysate meropenem levels exceeded the minimal inhibitory concentration of the pathogenic resistance organism (> 8 mg/L). Four patients responded to the treatment, whereas one developed treatment failure from fungal peritonitis.

**Data Availability Statement:** All relevant data are within the manuscript.

**Funding:** The author(s) received no specific funding for this work.

**Competing interests:** The authors have declared that no competing interests exist.

## Conclusion

An IP meropenem loading of 500 mg, followed by 125 mg/L every 6 hours, provided an adequate dialysate meropenem concentration and is an effective treatment for PD related peritonitis.

## Trial registration

Thai Clinical Trials Registry (TCTR20191121002) with date of first registration at 21/11/2019 (retrospectively registered).

## Introduction

Peritonitis, one of the most common complications of peritoneal dialysis (PD), leads to technical failure, increased hospitalization, and increased mortality among PD patients [1, 2]. It has been estimated that 20%–30% of PD-related peritonitis is caused by gram-negative organisms, which are associated with worse outcomes [3, 4]. Enterobacteriaceae and extended spectrum Beta-lactamases (ESBLs) producing bacteria are becoming increasingly resistant to many antibiotics [5, 6]. PD related peritonitis involving ESBL-producing gram-negative strains have demonstrated poorer clinical outcomes, including an increased risk of treatment failure [7]. ESBL-producing strains are usually susceptible to carbapenems and aminoglycosides. The rising number of multidrug-resistant organisms is resulting in the growing use of meropenem [8]. The International Society for Peritoneal Dialysis (ISPD) has recommended the intraperitoneal (IP) antibiotic route for PD-related peritonitis in view of the higher drug concentrations that are made available at the infection site than with the intravenous route [1]. Data relating to IP meropenem dosing is still limited to only a few studies. The recommended dosage for use in continuous ambulatory peritoneal dialysis (CAPD) is a once-daily IP infusion of 1 g of meropenem, which is allowed to dwell for at least 6 hours [1]. Nevertheless, this recommended dose is based on the pharmacokinetics performed in only one patient with *Enterobacter cloacae* peritonitis. The mentioned research demonstrated that an IP meropenem dose of 1 g administered once daily achieved meropenem plasma levels exceeding the minimal inhibitory concentration close to 100% of the time (T > MIC). However, no data for the dialysate drug concentration was reported [9]. Another study by Wiesholzer and colleagues investigated pharmacokinetic data relating to a single dose of meropenem in automated peritoneal dialysis patients without peritonitis [10], with regimen of 500 mg of IP meropenem, which was allowed to dwell for 15 hours. A mean target value of 40% T > MIC (with a European Committee on Antimicrobial Susceptibility Testing susceptibility breakpoint of 2 mg/L) was achieved in serum and dialysate after the IP administrations [10]. Nonetheless, the drug-dosing regimen utilized for automated peritoneal dialysis may not be applicable to CAPD, given the differences in their dialysate volumes and dwell times. Additionally, as an inflammeed peritoneum during the course of infection is known to alter peritoneal transport rate, the pharmacokinetics profiles of meropenem during active infection remains to be elucidated as prior studies has limited data on patients with active peritonitis [11].

IP antibiotics can be given as continuous (every exchange) or intermittent (once-daily) doses, with each achieving similar outcomes in terms of treatment success and relapse rates [12]. However, earlier studies have demonstrated that the intermittent IP administration of some beta-lactam antibiotics (such as cephalothin and ceftazidime) was associated with

subtherapeutic drug levels in both plasma and dialysate, causing concern for possible treatment failure [13, 14]. Similarly, the efficacy of meropenem depends on achieving an adequate %T > MIC (a bactericidal target of ~ 40%T > MIC) [15]. The concentration of meropenem should be maintained at two to four times that of MIC throughout the dosing interval. The Clinical and Laboratory Standards Institute determined that the MIC breakpoints of meropenem against Enterobacteriaceae are > 2 mg/L for susceptible microbes, 4 mg/L for intermediate microbes, and ≥ 8 mg/L for resistant microbes [16].

Therefore, a continuous administration of IP meropenem may provide a more adequate %T > MIC than an intermittent regimen. However, pharmacokinetic data for continuous IP meropenem administrations are still limited. There has only been a single case report which involved a CAPD patient with ESBL-producing *Escherichia coli* peritonitis. The patient received a continuous IP administration of meropenem at 125 mg/L for four cycles daily. The result showed plasma meropenem concentrations at steady-state and the mean dialysate at higher concentration than the MIC for the resistant organisms [17]. The study, however was done in only one patient and IP meropenem was administered after IP and IV routes of other antimicrobial. This study aimed to investigate the feasibility of a continuous IP application in providing an adequate level of plasma and dialysate meropenem in CAPD patients with peritonitis.

## Materials and methods

This prospective, descriptive, single-center study was performed in seven CAPD patients diagnosed with PD related peritonitis (Fig 1) at Siriraj Hospital, Mahidol University, from October 2017 to January 2018, in accordance with the International Conference on Harmonization Good Clinical Practice guidelines and the Declaration of Helsinki. Ethics approval was obtained from the Siriraj Institutional Review Board, Faculty of Medicine, Siriraj Hospital, Mahidol University (EC number Si 612/2017). At the time of submission of the study protocol, the Ethics Committee did not require registration for consideration of concept studies. This study was registered with the Thai Clinical Trials Registry (TCTR20191121002) with date of first registration at 21/11/2019. Written informed consent was obtained from all patients.

The inclusion criteria was for patients aged 18 years or older and have achieved stable CAPD for at least one month prior to enrolment. All patients were diagnosed with PD related peritonitis as per the ISPD guidelines of 2016, which required at least two of the following criteria: clinical features consistent with peritonitis (i.e., abdominal pain and a cloudy dialysis effluent); a dialysis-effluent white blood cell count of > 100/mm3, with more than 50% polymorph nuclear leukocytes (after a dwell time of at least 2 hours); and a positive dialysis-effluent culture(1). The exclusion criteria was history of a previous antibiotic treatment during the 30 days preceding the provisional enrolment; advanced cirrhosis (Child-Pugh class C); a concomitant exit site or tunnel infection; peritonitis from other known causes; presence of systemic inflammatory response syndrome; hemodynamic instability; concomitant antibiotic treatment for other conditions; and hypersensitivity or allergy to meropenem.

The primary outcomes were plasma meropenem levels, dialysate meropenem levels, and area under the curve (AUC) after continuous regimen of intraperitoneal meropenem administration in patients with PD related peritonitis. The secondary outcome was the side effects of the studied regimen.

### Study protocol

Initially, complete drainage of the peritoneal dialysate was performed in each patient. Subsequently, 2 L of dextrose dialysate were dwelled for 6 hours per exchange, with four exchanges

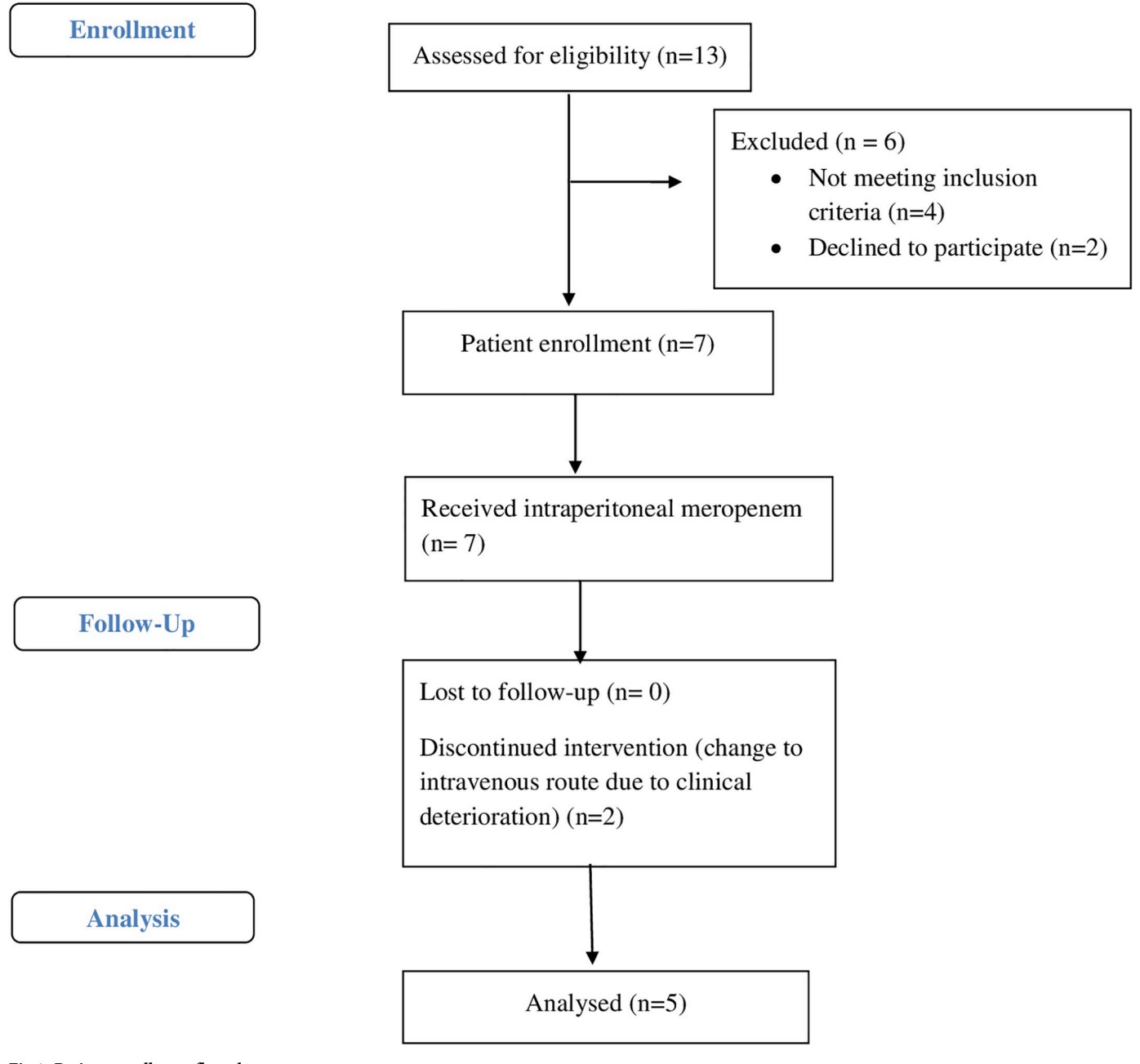

**Fig 1. Patient enrollment flow chart.**

daily. Previous research from Fitjer et al. demonstrated that IP meropenem 125 mg/L for 4 exchanges daily provided adequate plasma and dialysate meropenem level and good tolerability profile [17], and the ISPD guidelines for peritonitis recommended loading dose in continuous regimen of IP route to achieve therapeutic drug level [1]. Therefore, we selected to use loading dose of 500 mg meropenem for initial dose and maintainance with 125 mg/L of meropenem. All patients received an IP meropenem loading dose of 500 mg in the first bag, followed by maintenance therapy of 125 mg/L of IP meropenem in each exchange for four cycles daily by PD nurse (Fig 2). The average dextrose concentration during the treatment ranged

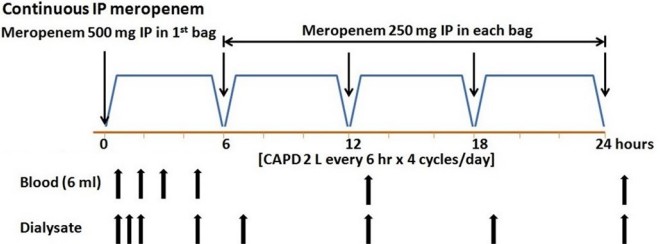

**Fig 2. Schematic diagram of venous blood sampling and dialysate meropenem concentrations.**

from 1.5% to 4.25%, with the dextrose levels being individualized by peritoneal ultrafiltration, residual urine, and volume status.

The baseline demographic data collected consisted of age, gender, body weight, height, body mass index, body surface area, etiology of end-stage renal disease, comorbidities, peritoneal dialysis vintage, and current medications.

In the case of patients with a residual renal function, the urine volume and urine concentration were measured from 24 hours of urine samples. The residual glomerular filtration rate was determined using the means of the urine creatinine clearance and the urine urea clearance, as follows:

$$\text{Creatinine clearance(ml/min)} = (\text{urine creatinine[mg/dl]} \times 24 - \text{hour urine volume[ml]})/(\text{serum creatinine[mg/dl]} \times 1440) \quad (1)$$

$$\text{Urea clearance(ml/min)} = (\text{urine urea [mg/dl]} \times 24 - \text{hour urine volume [ml]})/(\text{blood urea nitrogen[mg/dl]} \times 1440) \quad (2)$$

Before the administration of the IP meropenem, blood and dialysate samples were obtained as a baseline to verify the absence of any analytical interference from endogenous or exogenous sources. Time zero (zero hour) was defined as the time after complete drainage of the dialysate and before the meropenem administration. The blood samples (6 ml) were collected in lithium-heparin tubes for measurements of the plasma meropenem levels at baseline and at 1, 2, 4, 12, and 24 hours following the first dose of meropenem administration. The dialysate samples (10 ml) were collected to determine the meropenem levels at baseline, at 30 minutes, and at 1, 4, 6, 12, 18, and 24 hours after the first meropenem application (Fig 1). The dialysate samples were obtained by draining 200 ml of dialysate into the empty bag attached to the patient before aspirating 10 ml of dialysate for analysis. The remaining dialysate was readministered into the peritoneal cavity. The total volume of each drained dialysate was recorded.

All patients were followed up on Days 2, 5, and 14 after treatment. Dialysate samples were sent for cell count, differential count, and culture to evaluate the treatment response. Successful treatment was defined as improved clinical features of the peritonitis, coupled with a dialysis-effluent white blood cell count of less than 100/mm3—with less than 50% polymorph nuclear leukocytes—within five days. The antibiotics and treatment durations were adjusted according to the culture results. The antibiotic regimen was adjusted depending on the physician's decision. If those results were negative, the study drug was continued for 14 treatment days. All patients were followed up again on Day 14 after finishing the treatment to further evaluate their clinical responses.

The meropenem levels of the plasma and dialysate samples were quantified using high-performance liquid chromatography (HPLC). All HPLC analyses were carried out by a Waters Alliance liquid chromatography system (Waters Corporation, Milford, MA, USA), comprised of a Model 2695 Separate Module and a Model 2487 Dual Wavelength UV detector. HPLC separation was performed on a Symmetry C18 analytical column (5 μm, 250 x 4.6 mm I.D.), preceded by a sentry guard column C18 (5 μm, 20 x 3.9 mm I.D.; Waters Corporation, Milford, MA, USA), at a column temperature of 37°C (Temperature Control System, Waters Corporation, Milford, MA, USA). The mobile phase consisted of 0.5% of tetrabutylammonium hydroxide solution (25% in water), acetonitrile, and methanol (75:15:10, v/v), adjusted to pH 7.5 ± 0.1 with phosphoric acid. The mobile phase was filtered through a 0.2-μm membrane and degassed prior to use. The UV detection wavelength was 300 nm, and the flow rate was 1.3 mL/min. The analysis time was set at 10 min per sample, and the injection volume was 30 μL. The EmpowerPro software (Waters Corporation, Milford, MA, USA) was used to generate the standard curve by plotting areas under the curve of the extracted spike plasma versus various concentrations of meropenem [18].

The HPLC method was validated for its linearity, precision, accuracy, the limit of detection, and limit of quantitation, as per the International Conference on Harmonization guidelines. The meropenem calibration curve concentration range was 0.5–100 μg/mL. The linear equation of the calibration curve was $y = 114.67x - 4.0454$, and the linear correlation coefficients were 0.989 and 0.992 for plasma and peritoneal fluid, respectively. The intraday and interday precisions of meropenem were evaluated using three concentrations (1, 10, and 50 μg/mL), and the coefficient of variation (%CV) was lower than 2%. The method accuracy was determined using a recovery study conducted at three different concentrations (1, 10, and 50 μg/mL), and the average recoveries were 99.7 ± 1.2% and 99.3 ± 2.8% for the plasma and peritoneal fluid, respectively. In addition, the limits of detection for meropenem chromatographic determination were 0.5 and 1 μg/mL for the plasma and peritoneal fluid, respectively. Finally, the limits of quantitation for meropenem chromatographic determination were 1 and 2 μg/mL for the plasma and peritoneal fluid, respectively [18].

The blood samples (6 mL at each time point) were collected in the lithium-heparin tubes and centrifuged at 3,500 rpm for 10 min at 4°C. The blood cells were discharged, and the plasma samples were collected. Thereafter, the plasma and dialysate samples were divided into two aliquots of approximately 1 mL each before storing at -80°C for pending analysis. The meropenem concentrations of the plasma and dialysate were established using the HPLC method. A 400-μL aliquot of thawed clinical peritoneal fluid or plasma sample was transferred to a Nanosep 10K centrifugal filter device (Pall Life Sciences, Ann Arbor, MI, USA) and then centrifuged at 12,000 ×g for 10 min at room temperature. After the centrifugation, ~100 μL of the filtrate was collected in the filtrate reservoir of the device, and 30 μL of the supernatant was subsequently injected into the HPLC machine. The calibration curve of the meropenem was used to calculate the meropenem concentrations in the samples from their areas under the curve [18].

## Statistical analysis

For baseline characteristic of the included patients, categorical variables were presented as frequency (percentage), and continuous variables were presented as mean ± standard deviation (SD) for normal distributed data and median (interquartile range, IQR) for non-normal distributed data. For outcomes including the plasma meropenem levels, dialysate meropenem levels, and AUC were presented as mean ± SD and median (IQR). Statistical analysis was

**Table 1. Patient baseline demographic data.**

| Patient No. | Age (yr.) | Sex | BW (kg) | Height (cm) | BSA (m²) | Duration of PD (month) | 24 hr urine eGFR (ml/min) | Urine volume (ml/24 h) |
|---|---|---|---|---|---|---|---|---|
| 1 | 60 | M | 88 | 181 | 2.1 | 44 | 2.51 | 1,000 |
| 2 | 64 | F | 70 | 160 | 1.7 | 16 | 1.03 | 700 |
| 3 | 50 | M | 60 | 160 | 1.6 | 7 | 3.17 | 200 |
| 4 | 53 | F | 46 | 150 | 1.4 | 2.5 | 0 | 0 |
| 5 | 68 | F | 81 | 163 | 1.9 | 15 | 1.9 | 200 |

*BW, body weight; BSA, body surface area; ESRD, end-stage renal disease; PD, peritoneal dialysis; eGFR, estimated glomerular filtration rate.

performed using SPSS software version 18. The detail of sample size calculation is shown in a (S1 File) [10].

## Results

Seven PD patients with PD related peritonitis were enrolled in the study. Two patients were excluded. One patient developed clinical of bacteremia with hypotension at 2 hour after loading dose of IP meropenem and needed to switch to intravenous antibiotic. The other patient was found to have mixed organisms from PD fluid gram stain and secondary peritonitis was diagnosed. The baseline demographic data for all five patients are detailed in Table 1. The mean age of the patients was 59 (SD ± 7.5) years. The mean body weight was 69 (SD ± 16.7) kg, and the mean body surface area was 1.74 (SD ± 0.3) m². The most common cause of the end-stage renal disease was diabetic nephropathy (57.1%). The mean peritoneal dialysis vintage was 16.5 months, with a range of 2.5 to 44 months. The daily urine volumes ranged from 0 to 1,000 ml. The median estimated glomerular filtration rate from 24 hours of urine samples was 1.9 (1.03–2.51) ml/min.

### Plasma and dialysate meropenem concentrations

The median (IQR) and mean (± SD) for $AUC_{0-24}$ of meropenem in plasma were 739.75 (428.58–783.89) mg·h/liter and 608.85 (SD ± 390.30) mg·h/liter, respectively. In addition, the values in dialysate were 1215.19 (715.55–2506.90) mg·h/liter and 1687.08 (SD ± 1211.15) mg·h/liter, respectively.

The serial means and individual dialysate meropenem concentrations during the 24 hours of continuous IP meropenem regime are illustrated in Fig 3A and 3B. The highest mean of dialysate meropenem concentration was 158.1 (SD ± 62.9) mg/L, which was reached at 30 minutes. The median dialysate meropenem level was the lowest after 24 hours (37.689, IQR 19.65–40.65 mg/L). The dialysate meropenem levels from the continuous IP applications were well above the MIC of the pathogenic resistance organism (MIC > 8 mg/L) at every time point for all patients in the group.

The median plasma meropenem level was lowest at 1 hour (15.13, IQR 12.65–29.89 mg/L) and reached its peak at 2 hours (28.96, IQR 18.69–35.89 mg/L) after continuous IP meropenem administration. The mean and individual plasma meropenem levels after the continuous IP meropenem administration is presented in Fig 4A and 4B. The plasma meropenem levels of one patient were below the MIC (0.8–1.9 mg/L) throughout the 24 hours (Fig 4B). No adverse event was reported during the study period.

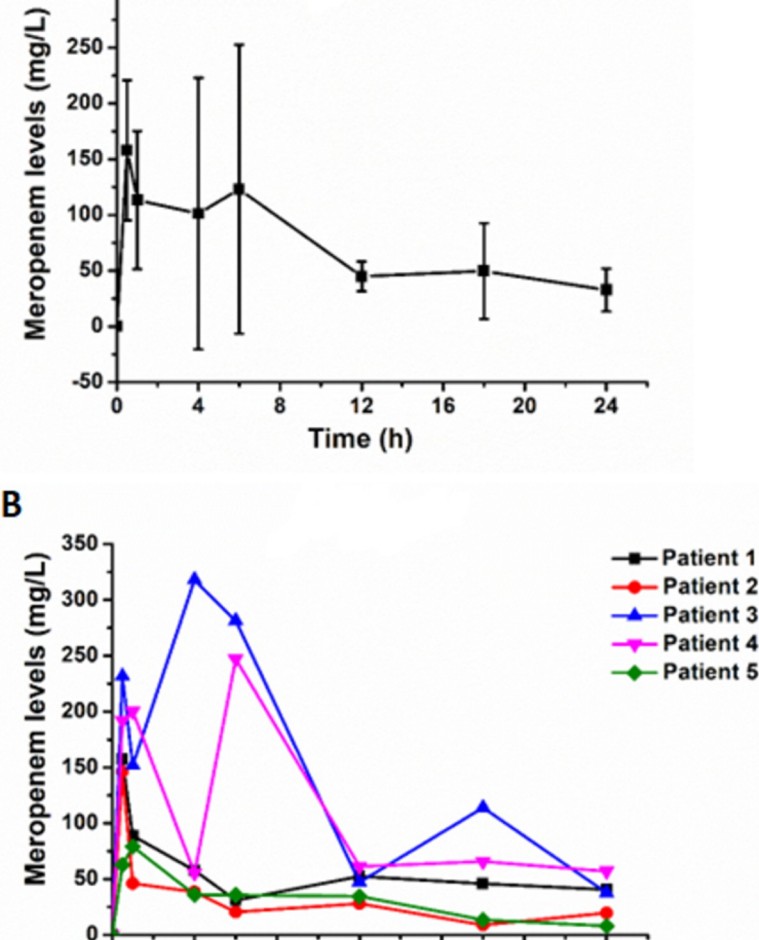

**Fig 3.** A: Dialysate meropenem levels after IP meropenem administration (mean ± SD). B: Individual dialysate meropenem levels after IP meropenem administration.

### Dialysate microbiological data and treatment responses

The microbiological causes of peritonitis are listed in Table 2. The causative organisms from the dialysate cultures were determined to be one case of *Streptococcus* group D, one of *Klebsiella pneumoniae*, and one of yeast; the remaining two patients had culture-negative peritonitis. Four patients who received continuous IP meropenem responded to the treatment. The remaining patient developed treatment failure from fungal peritonitis.

## Discussion

Meropenem is a beta-lactam antibiotic, with low to moderate volume of distributions and is low protein binding. Meropenem molecules are small and hydrophilic. Since meropenem is primarily eliminated in the urine, accumulation of the drug was seen in patients with renal impairment [19]. Previous report from Wiesholzer et al. demonstrated that IP meropenem

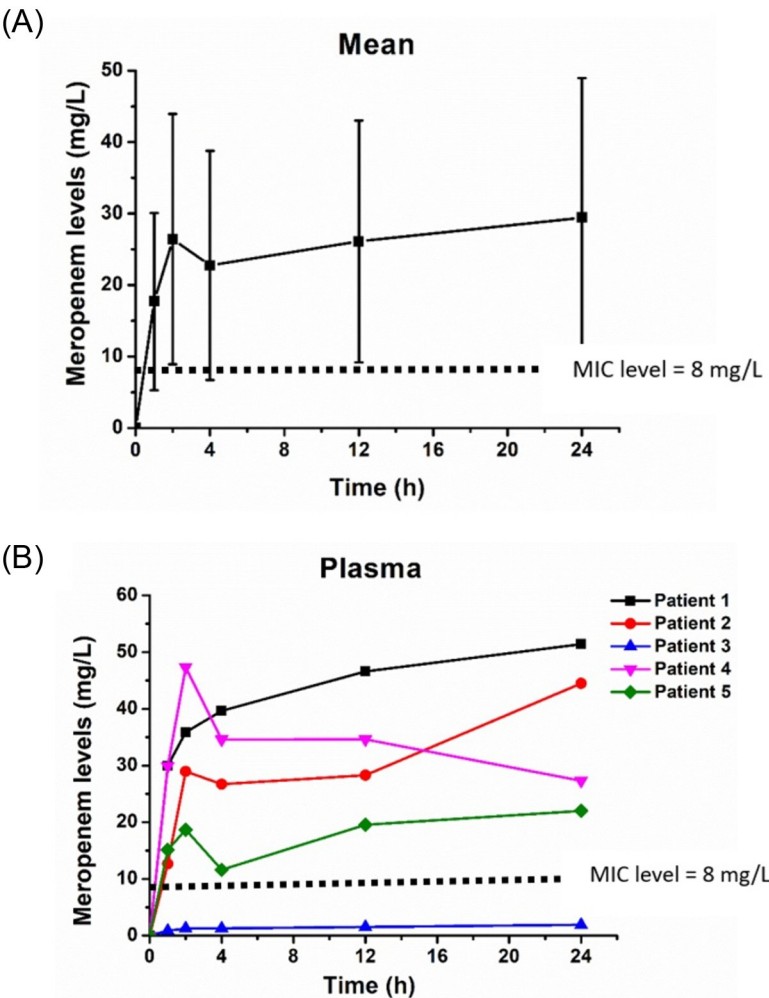

**Fig 4.** A: Plasma meropenem levels after IP meropenem administration (mean ± SD). B: Individual plasma meropenem levels after IP meropenem administration.

provided adequate dialysate drug level compared to intravenous route [10]. PD related peritonitis treatment required adequate antimicrobial dosing at site of infection, so IP route of meropenem is feasible and is the preferred route for PD related peritonitis from ESBL producing gram negative bateria. The available data on the usage of IP meropenem for PD-related peritonitis are limited to a few case reports for both the intermittent and continuous IP regimens.

**Table 2. Microbiological data and treatment responses.**

| Patient no. | Dialysate culture | Antibiotic treatment | Treatment responses | |
|---|---|---|---|---|
| | | | Day 5 | Day 14 |
| 1 | *Klebsiella pneumoniae* non-ESBL producing | Meropenem IP for 5 days, followed by ceftazidime IP for 16 days | Response | Response |
| 2 | No growth | Meropenem IP for 5 days, followed by ceftazidime and cefazolin for 9 days | Response | Response |
| 3 | *Streptococcus* group D | Meropenem IP for 14 days | Response | Response |
| 4 | Yeast (*Candida tropicalis*) | Meropenem IP for 5 days | Failure | – |
| 5 | No growth | Meropenem IP for 14 days | Response | Response |

This study has documented that a continuous IP meropenem regimen entailing a 500 mg IP loading followed by 125 mg/L IP every 6 hours can provide an adequate dialysate drug level throughout 24 hours. This finding is significant as the efficacy of meropenem is related to the percentage of the dosing interval during which the drug concentration exceeds the MICs of the pathogenic organisms [15]. In this study, the mean dialysate meropenem levels after continuous IP meropenem administration ranged from 32.6 to 158.1 mg/L over 24 hours, which provided an adequate meropenem level for Enterobacteriaceae with a MIC up to 8 mg/L (MIC $\leq$ 2 mg/L for susceptible organisms, and MIC > 8 mg/L for resistant organisms) [16]. This result is in line with the current ISPD recommendation, which prefers the use of IP antibiotics for PD-related peritonitis due to the high drug concentrations that are achieved at infection sites [1].

With the continuous IP meropenem group, the mean plasma meropenem levels exceeded the MIC within 1 hour and peaked at 24 hours after the first dose. Peritoneal membrane transport is higher during peritonitis, leading to an increase in meropenem bioavailability. Therefore, the continuous IP dose was effective as a treatment for PD related peritonitis.

Plasma meropenem levels below the MIC were observed in one patient (0.8–1.9 mg/L). However, for the treatment of PD-related peritonitis, dialysate meropenem levels are more critical than plasma meropenem levels in view of the need to achieve high drug levels at infection sites. Moreover, patients with suspected bacteremia were excluded from our study as the IP administration of antibiotics is not the preferred route for patients with bacteremia [1]. Therefore, the present regimen should be restricted in patients with PD related peritonitis who show no signs and no symptoms of systemic bacteremia.

The discrepancies in meropenem level between individuals can be explained by interpatient variability in the bioavailability of the IP meropenem. Thalhammer and Horl demonstrated significant interpatient variations in meropenem elimination among dialysis patients [19]. The other explanations were different residual renal function, peritoneal solute transport rate and perioneal surface area [20]. Whitty R. et al. reported treatment failure in PD related peritonitis was increased in patients with urinary creatinine clearance more than 5 mL/min regarding to inadequate drug level [21]. Previous study from Heimbürger O et al. demonstrated that the diffusive transport of small solutes through the peritoneal membrane is not influenced by the initial concentration of glucose [22].

Our results were similar to a case study by de Fitjer et al., which reported plasma and dialysate meropenem levels above MIC breakpoints using a continuous IP meropenem regimen (125 mg/L of meropenem, with four exchanges per day) in a CAPD-related peritonitis patient [17]. As our study has included a loading dose of meropenem, it is expected that the dialysate meropenem levels from our study to be significantly higher than the drug levels of the de Fitjer study, whose regime did not include a loading dose. Given that the drug level is much more than the usual MIC, though not at the toxicity level, further study could consider lowering the loading dose to minimize the possible side effect of high drug level as well as reducing the risk of inadequate dosing.

In another study by Vlaar et al., the plasma meropenem levels were measured after administering 1 gram of IP meropenem once daily in a patient with PD-related peritonitis. The peak plasma meropenem level was observed 4 hours after the IP administration, and the levels were well above the MIC breakpoints (2 mg/L) over 24 hours. However, the dialysate meropenem concentration was not measured by the study [9].

In addition, there have been reports of subtherapeutic dialysate drug levels arising from the once-daily IP dosing of other beta-lactam antibiotics, such as ceftazidime and cephalothin [13, 14]. A continuous IP regimen of beta-lactam antibiotics may be more effective than an intermittent regimen in providing adequate drug levels over 24 hours. Although, this study did not

measure drug level data after 24 hours, the IP meropenem was still added at the same 125 mg/L every 6 hours in continuous regimen. Therefore, the dialysate meropenem level would be adequately maintained.

Four patients (80%) responded to the continuous IP meropenem within five days. Another patient (20%) developed treatment failure due to fungal peritonitis. Thus shows that the causes of treatment failure in PD related peritonitis that may not be relate to the efficacy of the antibiotics used.

Although dialysis patients have an increased risk of inappropriate dosing and side effects from meropenem usage, our study found that continuous application of IP meropenem was well tolerated and safe. Furthermore, there were no signs of over or underdosing.

There are some limitations to our study. Firstly, the etiologic organisms of the PD-related peritonitis in all of our patients were non-ESBL-producing organisms (half of the patients were culture-negative). However, the objective of our study was to evaluate plasma and dialysate meropenem level from the continuous IP meropenem during peritonitis, regardless of the type of the organism identified. Our results demonstrated that continuous IP meropenem provided adequate dialysate meropenem levels to treat organisms up to MIC 8 mg/L, including ESBL-producing pathogens. This information has provided foundational safety data to rationalize specific enrollment of patients with ESBL-producing pathogens, which T > MIC is deemed critical for efficacy. Further studies then can be performed to explore the related pharmacokinetics as well as treatment outcomes in PD related peritonitis from ESBL-producing pathogens. Secondly, there was no adjustment according to residual renal function because of the very low residual glomerular filtration rates. For clinical applications in patients with residual renal function, plasma and dialysate meropenem levels may be affected and meropenem level in urine should be evaluated. Thirdly, we have not performed a full pharmacokinetics profiling in this study, however, we believed that the achieved blood level is likely reflecting the parameter that is generally used in clinical practice for the care of patients with infected peritonitis.

## Conclusions

An IP meropenem loading of 500 mg, which is continued by 125 mg/L every 6 hours, provides an adequate dialysate meropenem concentration. It could therefore be considered as an effective treatment for patients with PD related peritonitis.

## Supporting information

**S1 File. Sample size calculation.**
(DOCX)

**S2 File.**
(DOCX)

**S1 Checklist. TREND statement checklist.**
(PDF)

## Acknowledgments

This study was supported by the Routine-to-Research Unit, Siriraj Hospital, Mahidol University, Thailand.

## Author Contributions

**Conceptualization:** Thatsaphan Srithongkul, Sukit Raksasuk, Suchai Sritippayawan, Pornpan Koomanachai.

**Data curation:** Thatsaphan Srithongkul, Sukit Raksasuk, Bulaporn Techajongnumchai, Suchai Sritippayawan.

**Formal analysis:** Thatsaphan Srithongkul, Sukit Raksasuk, Bulaporn Techajongnumchai.

**Investigation:** Thatsaphan Srithongkul, Bulaporn Techajongnumchai.

**Methodology:** Thatsaphan Srithongkul, Bulaporn Techajongnumchai, Pornpan Koomanachai.

**Supervision:** Pornpan Koomanachai.

**Visualization:** Suchai Sritippayawan.

**Writing – original draft:** Thatsaphan Srithongkul.

**Writing – review & editing:** Thatsaphan Srithongkul, Sukit Raksasuk, Pornpan Koomanachai.

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
