## [Decision Letter · Decision Letter 0]

20 Dec 2023

PONE-D-23-31050Dialysate and Plasma Meropenem Concentrations in Continuous Intraperitoneal Regimen during Peritoneal-Dialysis-Related PeritonitisPLOS ONE

Dear Dr. Koomanachai,

Thank you for submitting your manuscript to PLOS ONE. After careful consideration, we feel that it has merit but does not fully meet PLOS ONE’s publication criteria as it currently stands. Therefore, we invite you to submit a revised version of the manuscript that addresses the points raised during the review process.

We look forward to receiving your revised manuscript.

Kind regards,

Christopher James Doig, MD MSc MA

Academic Editor

PLOS ONE

2. We note that you have selected “Clinical Trial” as your article type. PLOS ONE requires that all clinical trials are registered in an appropriate registry (the WHO list of approved registries is at https://www.who.int/clinical-trials-registry-platform/network/primary-registries
https://www.who.int/clinical-trials-registry-platform/network/primary-registries" https://www.who.int/clinical-trials-registry-platform/network/primary-registries
https://www.who.int/clinical-trials-registry-platform/network/primary-registries and more information on trial registration is at http://www.icmje.org/about-icmje/faqs/clinical-trials-registration/). Please state the name of the registry and the registration number (e.g. ISRCTN or ClinicalTrials.gov) in the submission data and on the title page of your manuscript. a) Please provide the complete date range for participant recruitment and follow-up in the methods section of your manuscript. b) If you have not yet registered your trial in an appropriate registry, we now require you to do so and will need confirmation of the trial registry number before we can pass your paper to the next stage of review. Please include in the Methods section of your paper your reasons for not registering this study before enrolment of participants started. Please confirm that all related trials are registered by stating: “The authors confirm that all ongoing and related trials for this drug/intervention are registered”. Please see http://journals.plos.org/plosone/s/submission-guidelines#loc-clinical-trials for our policies on clinical trials.

3. We note that your Data Availability Statement is currently as follows: [All relevant data are within the manuscript]

Additional Editor Comments:

Thank you for your contributions to the journal. Please find attached comments from individual reviewers.

It would be appropriate for you to respond to individual reviewer comments. There are also additional major comments included below which require a response. We look forward to receiving your revisions.

1. Within the introduction, there is a need to further justify the dosing regimen chosen. Was this based on general practice (e.g. usual dosing at your centre) or is there other data relied upon to determine the dosing regimen?

2. Further explanation of the 24h dialysate levels are required. It would appear from the data presented that the mean (SD) at 24h was 32.6 (19): this dispersion would suggest that there is not confidence that the mean would consistently be 4x greater than MIC's for resistant organisms (e.g. >8 mg/L).

3. Please provide further comment on the excretion pharmacokinetics of meropenem given that the plasma levels at 24h were still increasing in 3/5 participants (Figure 4B). The statement that evidence of adverse effects related to meropenem should be removed given the short duration of this study, and the study did not include a protocol for examining treatment related adverse effects.

4. The discussion should be more specific that the potential use of IP meropenem with this dosing regimen would be restricted to individuals without bacteremia, or evidence of (severe) sepsis. The term 'septicemia' should be avoided.

Reviewers' comments:

Reviewer's Responses to Questions

**Comments to the Author**

1. Is the manuscript technically sound, and do the data support the conclusions?

Reviewer #1: Yes

Reviewer #2: Yes

Reviewer #3: Yes

Reviewer #4: Partly

2. Has the statistical analysis been performed appropriately and rigorously? 

Reviewer #1: No

Reviewer #2: Yes

Reviewer #3: Yes

Reviewer #4: Yes

3. Have the authors made all data underlying the findings in their manuscript fully available?

Reviewer #1: Yes

Reviewer #2: Yes

Reviewer #3: Yes

Reviewer #4: Yes

4. Is the manuscript presented in an intelligible fashion and written in standard English?

Reviewer #1: No

Reviewer #2: Yes

Reviewer #3: Yes

Reviewer #4: Yes

5. Review Comments to the Author

Reviewer #1: The authors here justify their work on PD for 5 patients treated with IP meropenem in a largely descriptive study.

As this isn't a randomised trial figure 1 could usefully just be called a flowchart as CONSORT is for randomised trials.

The sample size needs to be explained more fully - the first sentence I think is missing a word such as estimate. A sample size like 6 is too small for the standard sample size formulae as the approximations are not good for small numbers - as such please give more details on the calculation. Is the "+20% dropoff" an increase of 20% to allow for 16% dropout or an inflation of 25% to allow for 20% dropout? Are the numbers in this section with +/- symbols sd or sem?

Please remember that for normality double the standard deviation needs to be less than the mean. As such the reporting of mean and sd is not always a fair measure of the distribution.

At what point did the 2 patients drop out - weer any post entry datapoints collected? How did these look as we know these are not responders and as such are particularly of interest.

The legend on figure 3a does not tell us what the error bard represent. Ditto for figure 4a.

Reviewer #2: Thank you for asking me to review this well written manuscript. As the title describes to comments on dialysate and plasma concentrations of meropenem in patients on peritoneal dialysis using 5 subjects to do so.

Whilst it is a small series, this commentary is not on a common disease profile and it is important to document infection site antibiotic concentrations.

I have few comments.

Firstly I think the authors should clearly document how this manuscript adds to current literature that their reference 17 already describes.

I don’t see why the authors use a subheading in line 208 mentioning sample size calculation? The following description does not justify their sample size of five patients.

The authors statement in line 263 of “proven” should rather say “documents”…ie ……The present study has proven that a continuous IP meropenem regimen…I suggest ….The present study documents that a continuous IP meropenem regimen….

Ref 5 should have a journal abbreviation

Can the authors please comment on the second spike in levels of patient 3 and patient 4 in figure 3b

Reviewer #3: The authors report on the dialysate and plasma meropenem concentrations in CAPD patients.

1. The study consists of 5 patients, which is usually normal for a pharmocokinetic study. However, the patient population is far from uniform in this example. The majority of patients do not even have gram (-) bacteria. As the authors state, one has fungal, one has streptococci and the two are culture negative. This makes the response rates to meropenem irrelevant.

2. Even though there has been multiple blood and dialysate sampling, there is no PK data or compartment modeling using a software program incorporating the pharmocakinetic parameters.

3. According to this study, all the levels achieved are way over 8mg/L so all this study tells me is that the dose used in the study was too much.

I think the authors should use this study and data as a preliminary study and plan another one with a uniform patient population and different dosing regimen, incorporating the parameters into the system.

Reviewer #4: I wish to thank the authors for this submission on an important topic, that is treating resistant infections in PD peritonitis. This is important work, and appears to be a shortcoming in the existing literature.

1. Power calculation - you have performed a descriptive, proof of concept study and for this reason there is no existing literature with which to perform a power calculation. I also do not think it is necessary, although warrants toning down the assertion that this is an effective treatment for PD peritonitis. Your study demonstrates that therapeutic levels of meropenem can be achieved in the dialysate, and now further studies to assess the efficacy, for example in RCT, would be warranted.

2. Pharmacokinetics-it may be valuable to explain to the readers the pharmacokinetics of meropenem. This background would help explain why administration via the PD route is feasible, and I would suggest including parameters such as the volume of distribution, protein binding and route of excretion. It would also be helpful to explain any of the variables with peritoneal dialysis that affect pharmacokinetic properties such as absorption, distribution and excretion. With the discrepancies you see between individuals, it may be important to describe the characteristics of their dialysis. I wondered about the dextrose concentration and if this plays a role in the movement across the peritoneal membrane. Additionally, does residual renal function play a role? If meropenem undergoes tubular excretion, is all residual renal function equal?

3. Discussion - As mentioned previously, I think additional studies to demonstrate efficacy would be warranted, as you have shown what appears to be effective dialysate concentrations. Are there other variables that explain the differences in meropenem concentrations beyond inter-individual variability? It would be helpful to consider both the pharmacokinetic considerations described above, but also potential disease characteristics that play a role. For example patient 4, is there something unique about fungal peritonitis that makes it more permeable or pro inflammatory there by decreasing the concentration gradient observed and resulting in a more uniform concentration between the two compartments? Further to these pharmacokinetic considerations above, would you advocate for measuring meropenem concentrations in the urine for patients with residual renal function?

I hope you will find these comments

6. PLOS authors have the option to publish the peer review history of their article (what does this mean?). If published, this will include your full peer review and any attached files.

Reviewer #1: No

Reviewer #2: No

Reviewer #3: No

Reviewer #4: No

---

## [Author Response · Author response to Decision Letter 0]

18 Feb 2024

In response to Editor: 

1.Within the introduction, there is a need to further justify the dosing regimen chosen. Was this based on general practice (e.g. usual dosing at your center) or is there other data relied upon to determine the dosing regimen?

Thank you for your comment. According to previous research from Fitjer et al. demonstrated that IP meropenem 125 mg/L for 4 exchanges daily provided adequate plasma and dialysate meropenem level and safety profiles. According to ISPD guidelines for peritonitis recommended loading dose in continuous regimen of IP route to achieve therapeutic drug level, so we selected to use loading dose of 500 mg meropenem for initial dose. This explanation has been added in text. 

2.Further explanation of the 24h dialysate levels are required. It would appear from the data presented that the mean (SD) at 24h was 32.6 (19): this dispersion would suggest that there is not confidence that the mean would consistently be 4x greater than MIC's for resistant organisms (e.g. >8 mg/L).

Thank you for your suggestion. Unfortunately, we did not have the data of drug level after 24 hours. However, the meropenem IP will be added 125 mg/L every 6 hours in continuous IP regimen, that will maintain the meropenem level in dialysate. Furthermore, we did not expect to use meropenem to treat CRE (MIC >8mg/L). If CRE was suspicious, we would rather use other ATB such as colistin. This text has been added in discussion section. 

3. Please provide further comment on the excretion pharmacokinetics of meropenem given that the plasma levels at 24 h were still increasing in 3/5 participants (Figure 4B). The statement that evidence of adverse effects related to meropenem should be removed given the short duration of this study, and the study did not include a protocol for examining treatment related adverse effects.

Thank you for your comment. The plasma level of meropenem at 24 hour was increased due to the third dose of IP meropenem at 18 hr. Similarly, P.J. Vlaar reported the highest concentration of plasma meropenem was observed after 4 hours of IP meropenem administration. 

The statement of adverse effects related to meropenem has been removed as suggestion. 

4. The discussion should be more specific that the potential use of IP meropenem with this dosing regimen would be restricted to individuals without bacteremia, or evidence of (severe) sepsis. The term 'septicemia' should be avoided.

Thank you very much. the text of “Therefore, the present regimen should be restricted in the patient with PD related peritonitis, who did not have signs and symptoms of systemic bacteremia” has been added in the text. 

In response to Reviewer 1:

1. As this isn't a randomised trial figure 1 could usefully just be called a flowchart as CONSORT is for randomised trials.

Thank you. The legend of Figure1 has been changed to “patient enrollment flow chart” in the text.

2. The sample size needs to be explained more fully - the first sentence I think is missing a word such as estimate. A sample size like 6 is too small for the standard sample size formulae as the approximations are not good for small numbers - as such please give more details on the calculation. Is the "+20% dropoff" an increase of 20% to allow for 16% dropout or an inflation of 25% to allow for 20% dropout? Are the numbers in this section with +/- symbols sd or sem?

Thank you for your suggestion. According to our study design, which was a descriptive study to evaluate the level of meropenem in dialysate, and for this reason there is no existing literature with which to perform a power calculation. Therefore, the section of power calculation was not necessary and has been removed in the text.

3. Please remember that for normality double the standard deviation needs to be less than the mean. As such the reporting of mean and sd is not always a fair measure of the distribution.

Thank you for your advice. Median and IQR has been added in the reporting of non-normal distribution data.

4. At what point did the 2 patients drop out - were any post entry datapoints collected? How did these look as we know these are not responders and as such are particularly of interest.

Thank you for your comment. One patient developed clinical of SIRS and hypotension at 2 hours after loading dose of intraperitoneal meropenem and another patient was diagnosed with secondary peritonitis after first dose of intraperitoneal meropenem, so the route of antimicrobial was changed to intravenous route. The explanation has been added in the text.

5. The legend on figure 3a does not tell us what the error bard represent. Ditto for figure 4a.

Thank you. The bared represent mean ± SD. The legend of figure 3a and 4a was revised in the text.

In response to Reviewer 2:

1.Firstly I think the authors should clearly document how this manuscript adds to current literature that their reference 17 already describes.

Thank you for your comment. According to previous study from Fitjer et al. demonstrated that IP meropenem 125 mg/L for 4 exchanges daily provided adequate plasma and dialysate meropenem level. However, the study was done in only one patient and IP meropenem was administered after intravenous antimicrobial. Therefore, our study aimed to evaluate meropenem level from continuous regimen of IP meropenem in more patients. The explanation has been added in the article.

2. I don’t see why the authors use a subheading in line 208 mentioning sample size calculation? The following description does not justify their sample size of five patients.

Thank you for your comments. According to the study design which was a descriptive study to evaluate the level of meropenem in dialysate, and for this reason there is no existing literature with which to perform a power calculation. Therefore, the section of power calculation was not necessary and has been removed in the text. 

3.The authors statement in line 263 of “proven” should rather say “documents”…ie ……The present study has proven that a continuous IP meropenem regimen…I suggest ….The present study documents that a continuous IP meropenem regimen….

Thank you for your suggestion. The statement has been changed follow your recommendation. 

4.Ref 5 should have a journal abbreviation

Thank you. The journal abbreviation has been replaced in reference 5.

5. Can the authors please comment on the second spike in levels of patient 3 and patient 4 in figure 3b

Thank you for your comment. The second spike in dialysate levels in patient 3 and 4 at 4 hours and 6 hours may be explained by directional of solute transport through peritoneal membrane that caused diffusion of drug from plasma to dialysate. 

In response to Reviewer 3:

The authors report on the dialysate and plasma meropenem concentrations in CAPD patients.

1. The study consists of 5 patients, which is usually normal for a pharmocokinetic study. However, the patient population is far from uniform in this example. The majority of patients do not even have gram (-) bacteria. As the authors state, one has fungal, one has streptococci and the two are culture negative. This makes the response rates to meropenem irrelevant.

Thank you for your comment. I agree that from our present article could not proved the efficacy of IP meropenem due to the population in our study did not have gram negative bacteremia. However, the primary objective of the study was to evaluated plasma and dialysate meropenem level from IP meropenem during peritonitis, which may interfere drug bioavailability and absorption. The further study is needed for proven efficacy of IP meropenem in PD related peritonitis from gram negative bacteria. This explanation has been stated in the text.

2. Even though there has been multiple blood and dialysate sampling, there is no PK data or compartment modeling using a software program incorporating the pharmacokinetic parameters.

Thank you for your comment. Unfortunately, we did not have pharmacokinetic data in our study. The further study with PK data is needed to evaluate efficacy of present regimen of meropenem. This text has been added in limitation section. 

3. According to this study, all the levels achieved are way over 8mg/L so all this study tells me is that the dose used in the study was too much.

I think the authors should use this study and data as a preliminary study and plan another one with a uniform patient population and different dosing regimen, incorporating the parameters into the system.

Thank you for your suggestion. I agree that further study is needed for evaluating pharmacokinetic and treatment outcome and the text has been added in the limitation of study. 

In response to Reviewer 4:

I wish to thank the authors for this submission on an important topic, that is treating resistant infections in PD peritonitis. This is important work, and appears to be a shortcoming in the existing literature.

1. Power calculation - you have performed a descriptive, proof of concept study and for this reason there is no existing literature with which to perform a power calculation. I also do not think it is necessary, although warrants toning down the assertion that this is an effective treatment for PD peritonitis. Your study demonstrates that therapeutic levels of meropenem can be achieved in the dialysate, and now further studies to assess the efficacy, for example in RCT, would be warranted.

Thank you for your suggestions. We agreed that power calculation did not necessary in our study that aimed to evaluate therapeutic levels of meropenem. Therefore, the section of power calculation has been removed in the text. 

2. Pharmacokinetics-it may be valuable to explain to the readers the pharmacokinetics of meropenem. This background would help explain why administration via the PD route is feasible, and I would suggest including parameters such as the volume of distribution, protein binding and route of excretion. It would also be helpful to explain any of the variables with peritoneal dialysis that affect pharmacokinetic properties such as absorption, distribution and excretion. With the discrepancies you see between individuals, it may be important to describe the characteristics of their dialysis. I wondered about the dextrose concentration and if this plays a role in the movement across the peritoneal membrane. Additionally, does residual renal function play a role? If meropenem undergoes tubular excretion, is all residual renal function equal?

Thank you for your suggestions. Meropenem is beta-lactam antibiotic, which demonstrated low to moderate volume of distributions and low protein binding. Meropenem molecules is small and hydrophilic. Since meropenem is primarily eliminated unchanged in the urine, accumulation of the drug was seen in patients with renal impairment. Previous report from Wiesholzer et al. demonstrated that IP meropenem provided adequate dialysate drug level better than intravenous route. For PD related peritonitis treatment required adequate antimicrobial dosing at site of infection, so IP route of meropenem is feasible and preferred in PD related peritonitis from ESBL producing gram negative bacteria. 

The other explanations for discrepancies between individuals, may be explained by different residual renal function, peritoneal solute transport rate and peritoneal surface area. Whitty R. et al. reported treatment failure was increased in patients with urinary creatinine clearance more than 5 mL/min. However, previous study from Heimbürger O et al. demonstrated that the diffusive transport of small solutes through the peritoneal membrane is not influenced by the initial concentration of glucose. The explanation has been added in the text. 

3. Discussion - As mentioned previously, I think additional studies to demonstrate efficacy would be warranted, as you have shown what appears to be effective dialysate concentrations. Are there other variables that explain the differences in meropenem concentrations beyond inter-individual variability? It would be helpful to consider both the pharmacokinetic considerations described above, but also potential disease characteristics that play a role. For example, patient 4, is there something unique about fungal peritonitis that makes it more permeable or pro inflammatory there by decreasing the concentration gradient observed and resulting in a more uniform concentration between the two compartments? Further to these pharmacokinetic considerations above, would you advocate for measuring meropenem concentrations in the urine for patients with residual renal function?

Thank you for your comments. Other factors that played a role of the difference in meropenem concentration would be different residual renal function, peritoneal membrane transport and peritoneal surface area. Patients with significant residual renal function, meropenem concentration in urine should be measured for pharmacokinetics consideration in the future research.

---

## [Decision Letter · Decision Letter 1]

10 Jun 2024

PONE-D-23-31050R1Dialysate and Plasma Meropenem Concentrations in Continuous Intraperitoneal Regimen during Peritoneal-Dialysis-Related PeritonitisPLOS ONE

Dear Dr. Koomanachai,

Thank you for submitting your manuscript to PLOS ONE. After careful consideration, we feel that it has merit but does not fully meet PLOS ONE’s publication criteria as it currently stands. Therefore, we invite you to submit a revised version of the manuscript that addresses the points raised during the review process.

We look forward to receiving your revised manuscript.

Kind regards,

Ankur Shah

Academic Editor

PLOS ONE

Journal Requirements:

Additional Editor Comments:

The authors are requested to respond to the concerns about the patient population including both sample size and recruitment as well as the heterogeneity, particularly in regards to response rates.

Reviewers' comments:

Reviewer's Responses to Questions

**Comments to the Author**

1. If the authors have adequately addressed your comments raised in a previous round of review and you feel that this manuscript is now acceptable for publication, you may indicate that here to bypass the “Comments to the Author” section, enter your conflict of interest statement in the “Confidential to Editor” section, and submit your "Accept" recommendation.

Reviewer #1: (No Response)

Reviewer #2: All comments have been addressed

Reviewer #3: (No Response)

Reviewer #4: All comments have been addressed

2. Is the manuscript technically sound, and do the data support the conclusions?

Reviewer #1: No

Reviewer #2: Yes

Reviewer #3: Partly

Reviewer #4: Yes

3. Has the statistical analysis been performed appropriately and rigorously? 

Reviewer #1: No

Reviewer #2: N/A

Reviewer #3: N/A

Reviewer #4: N/A

4. Have the authors made all data underlying the findings in their manuscript fully available?

Reviewer #1: Yes

Reviewer #2: Yes

Reviewer #3: Yes

Reviewer #4: Yes

5. Is the manuscript presented in an intelligible fashion and written in standard English?

Reviewer #1: Yes

Reviewer #2: Yes

Reviewer #3: Yes

Reviewer #4: No

6. Review Comments to the Author

Reviewer #1: Thank you for your response to my previous comments. I am now very confused. The sample size statement is what the original planned recruitment was - simply removing the numbers and the rationale makes it look like there was no planned recruitment which is in direct contradiction of the previous version of the manuscript. The sample size as this is a prospective study is required and a valid justification of it including dropout which was not handled clearly in the original version of the paper. It is not scientifically valid to remove this section as it now misleads the reader as to the planned intent

Reviewer #2: The authors have adequately addressed all the comments I had made. Whilst I think they have addressed other reviewer comments, the other reviewers should comment.

Reviewer #3: Unfortunately, my previous comments are still relevant such as the lack of a uniform study population as well as pharmacokinetic data and very high meropenem levels throughout the study. Even though the idea of this manuscript is sound and has merit, the study itself needs further work.

Reviewer #4: The introduction would benefit with a proof read by a native English speaker for flow and clarity. There are a few sentences that should be rewritten, for example "Enterobacteriaceae are increasingly becoming resistant to many antibiotics regarding extended-spectrum β-lactamases (ESBLs)producing [5, 6].

Additionally, I think it may be worth mentioning how much higher the levels are above the MIC targets and that future studies may include exploring dose reductions.

Line 88 - Should that line read > 2 mg/L for susceptible organisms?

Line 98 - I think this study examines feasibility, not efficacy, efficacy would need to be outcomes based.

Line 222/292 - septicaemia should be removed and alternate terminology used (suspected bacteremia).

7. PLOS authors have the option to publish the peer review history of their article (what does this mean?). If published, this will include your full peer review and any attached files.

Reviewer #1: No

Reviewer #2: No

Reviewer #3: No

Reviewer #4: No

---

## [Author Response · Author response to Decision Letter 1]

16 Jul 2024

Dear Reviewers,

Many thanks for your constructive feedback and suggestions. We have addressed all of the issues below and in the manuscript. We believe these changes have improved the manuscript considerably and hope that it is now acceptable for publication.

In response to Reviewer 1:

Reviewer #1: The sample size as this is a prospective study is required and a valid justification of it including dropout which was not handled clearly in the original version of the paper. It is not scientifically valid to remove this section as it now misleads the reader as to the planned intent.

Thank you for your comment. We acknowledged that sample size calculation and detail is important for most prospective studies. We have now added the sample size calculation in the supplement. There, we also addressed our rationale of the adjustment for sample size dropout as well. 

In response to Reviewer 2:

The authors have adequately addressed all the comments I had made. Whilst I think they have addressed other reviewer comments, the other reviewers should comment.

Thank you for your constructive feedback.

In response to Reviewer 3:

Unfortunately, my previous comments are still relevant such as the lack of a uniform study population as well as pharmacokinetic data and very high meropenem levels throughout the study. Even though the idea of this manuscript is sound and has merit, the study itself needs further work.

We appreciate your feedback on the uniformity of the study population and the concern of the drug level. We have added our rationale in the discussion to further clarify this. As we aimed to study the drug level of meropenem in an inflamed peritoneum, it is difficult to rationalize this type of study in patients with proven ESBL-gram negative bacteria which requires that the drug level has to be achieved. This study result can then be used as a foundational data for further study with specific enrollment for patients with ESBL-gram negative bacteria.

In regards to the high drug level, it is a great point that you have commented. However, we have not observed any signs of meropenem toxicity in our patient. Acknowledging that the drug level is higher than that what is needed, we have added in the discussion to suggest a change in the loading regimen in future studies to both minimize the risk of inadequate dosing and toxicity.

Unfortunately, the full pharmacokinetics profiles of the drug is beyond the scope of our work and we are not able to profile such data at this time. We have added this limitation in our discussion as well.

In response to Reviewer 4:

1.The introduction would benefit with a proof read by a native English speaker for flow and clarity. There are a few sentences that should be rewritten, for example "Enterobacteriaceae are increasingly becoming resistant to many antibiotics regarding extended-spectrum β-lactamases (ESBLs)producing.

Changes has been made to as follow. Thank you for your feedback. We have also sent the manuscript to the language support service of Mahidol University to further adjust the flow of the wording used in this manuscript.

“Enterobacteriaceae and extended spectrum Beta-lactamases (ESBLs) producing bacteria are becoming increasingly resistant to many antibiotics[5, 6]. PD related peritonitis involving ESBL-producing gram-negative strains have demonstrated poorer clinical outcomes, including an increased risk of treatment failure [7].”

2.Additionally, I think it may be worth mentioning how much higher the levels are above the MIC targets and that future studies may include exploring dose reductions.

Thank you for pointing out on the aspect of the level of meropenem. To clearly show ‘how much higher’, we have adjusted figure 4B (blood meropenem level) to include the MIC level in dash line. We have also included a statement about dose reduction in future studies in the discussion. 

3. Line 88 - Should that line read > 2 mg/L for susceptible organisms?

Thank you so much for your hardwork here. We have adjusted accordingly.

4. Line 98 - I think this study examines feasibility, not efficacy, efficacy would need to be outcomes based.

We agree with your comment and has now adjusted the language to be a feasibility study.

Line 222/292 - septicaemia should be removed and alternate terminology used (suspected bacteremia).

Thank you for your suggestion. We have revised accordingly.

---

## [Decision Letter · Decision Letter 2]

15 Aug 2024

PONE-D-23-31050R2Dialysate and Plasma Meropenem Concentrations in Continuous Intraperitoneal Regimen during Peritoneal-Dialysis-Related PeritonitisPLOS ONE

Dear Dr. Koomanachai,

Thank you for submitting your manuscript to PLOS ONE. After careful consideration, we feel that it has merit but does not fully meet PLOS ONE’s publication criteria as it currently stands. Therefore, we invite you to submit a revised version of the manuscript that addresses the points raised during the review process.

We look forward to receiving your revised manuscript.

Kind regards,

Ankur Shah

Academic Editor

PLOS ONE

Journal Requirements:

**Additional Editor Comments:**

Please address the methodological concerns of reviewer 1, thank you.

Reviewers' comments:

Reviewer's Responses to Questions

**Comments to the Author**

1. If the authors have adequately addressed your comments raised in a previous round of review and you feel that this manuscript is now acceptable for publication, you may indicate that here to bypass the “Comments to the Author” section, enter your conflict of interest statement in the “Confidential to Editor” section, and submit your "Accept" recommendation.

Reviewer #1: (No Response)

Reviewer #3: All comments have been addressed

Reviewer #4: All comments have been addressed

2. Is the manuscript technically sound, and do the data support the conclusions?

Reviewer #1: (No Response)

Reviewer #3: Yes

Reviewer #4: Yes

3. Has the statistical analysis been performed appropriately and rigorously? 

Reviewer #1: (No Response)

Reviewer #3: N/A

Reviewer #4: N/A

4. Have the authors made all data underlying the findings in their manuscript fully available?

Reviewer #1: (No Response)

Reviewer #3: Yes

Reviewer #4: Yes

5. Is the manuscript presented in an intelligible fashion and written in standard English?

Reviewer #1: (No Response)

Reviewer #3: Yes

Reviewer #4: Yes

6. Review Comments to the Author

Reviewer #1: Thank you for your response - the primary outcome was not "to evaluate ..." - this is the primary aim. The primary outcome is the variable used to measure this .Please clarify and state in the methods that the aim was to estimate the parameter with a certain degree of accuracy. Please tie the variable used for the power calculation in with the numbers given in the sample size calculation which would appear to be AUC as opposed to the information given in the results.

The results likewise do not report a normal distribution which is what is assumed - and please justify an infinite population assumption for 6 patients.in a sample where one has to estimate the variance as well.

Reviewer #3: The authors have addressed and included the limitations of the study in the manuscript, pointing out that meropenem dosing in further studies would be adjusted to this study. This aim and understanding make the study more relevant.

Reviewer #4: Thank you for addressing all of my comments. I think this is important work and I appreciate that you have taken the time to address everything.

7. PLOS authors have the option to publish the peer review history of their article (what does this mean?). If published, this will include your full peer review and any attached files.

Reviewer #1: No

Reviewer #3: No

Reviewer #4: No

---

## [Author Response · Author response to Decision Letter 2]

3 Sep 2024

Dear Editor and Reviewers,

We do appreciate all of your suggestions and comments. We have addressed all of the issues below and in the manuscript. We believe these changes have improved the manuscript considerably and hope that it is now acceptable for publication.

In response to Reviewer 1:

1. The primary outcome was not "to evaluate ..." - this is the primary aim. The primary outcome is the variable used to measure this. Please clarify and state in the methods that the aim was to estimate the parameter with a certain degree of accuracy.

Thank you for your comment. We have adjusted in the main text as your suggestion which has provided more clarify on the definition of the primary outcome.

2. Please tie the variable used for the power calculation in with the numbers given in the sample size calculation which would appear to be AUC as opposed to the information given in the results.

Thank you for raising up this point on the sample calculation. We added the sample size calculation based on previously published AUC0–24 of meropenem in serum reported by Wiesholzer et al. in the figure attached below and in the supplementary file (S1 file). Additionally, we have included the outcome regard to AUC0–24 of meropenem in plasma and dialysate as shown in the revised manuscript. 

3. The results likewise do not report a normal distribution which is what is assumed 

We agree that it is not straight forward to assume normal distribution in this sample. Following your recommendation, now we have added both mean ± SD and medium + IQR to ensure all dimensions of the data are represented. 

4. Please justify an infinite population assumption for 6 patients in a sample where one has to estimate the variance as well.

Although we do not know the total target population, it is likely that the target population for our study is very large. Also, given the random sampling from a large pool of patients in our hospital, there should be a negligible sampling fraction. Given that prior study with similar goals has also provided the estimated variance, we felt that using the infinite population mean formula should be reasonable.

---

## [Decision Letter · Decision Letter 3]

2 Oct 2024

Dialysate and Plasma Meropenem Concentrations in Continuous Intraperitoneal Regimen during Peritoneal-Dialysis-Related Peritonitis

PONE-D-23-31050R3

Dear Dr. Koomanachai,

We’re pleased to inform you that your manuscript has been judged scientifically suitable for publication and will be formally accepted for publication once it meets all outstanding technical requirements.

Kind regards,

Ankur Shah

Academic Editor

PLOS ONE

Additional Editor Comments (optional):

All comments have been addressed, thank you

Reviewers' comments:

Reviewer's Responses to Questions

**Comments to the Author**

1. If the authors have adequately addressed your comments raised in a previous round of review and you feel that this manuscript is now acceptable for publication, you may indicate that here to bypass the “Comments to the Author” section, enter your conflict of interest statement in the “Confidential to Editor” section, and submit your "Accept" recommendation.

Reviewer #1: All comments have been addressed

2. Is the manuscript technically sound, and do the data support the conclusions?

Reviewer #1: (No Response)

3. Has the statistical analysis been performed appropriately and rigorously? 

Reviewer #1: (No Response)

4. Have the authors made all data underlying the findings in their manuscript fully available?

Reviewer #1: (No Response)

5. Is the manuscript presented in an intelligible fashion and written in standard English?

Reviewer #1: (No Response)

6. Review Comments to the Author

Reviewer #1: (No Response)

7. PLOS authors have the option to publish the peer review history of their article (what does this mean?). If published, this will include your full peer review and any attached files.

Reviewer #1: No

---

## [Editor Report · Acceptance letter]

29 Oct 2024

PONE-D-23-31050R3 

PLOS ONE

Dear Dr. Koomanachai, 

I'm pleased to inform you that your manuscript has been deemed suitable for publication in PLOS ONE. Congratulations! Your manuscript is now being handed over to our production team.

Kind regards, 

on behalf of

Dr. Ankur Shah 

Academic Editor

PLOS ONE